# DrivingRecon: Large 4D Gaussian Reconstruction Model For Autonomous Driving

**Hao LU**[1,2,3*]**, Tianshuo XU**[1,2]**, Wenzhao ZHENG**[3]**, Yunpeng ZHANG**[4]**, Wei ZHAN**[3]**,
Dalong DU**[4]**, Masayoshi Tomizuka**[3]**, Kurt Keutzer**[3]**, Yingcong CHEN**[1,2,†]

[1]The Hong Kong University of Science & Technology (Guangzhou),
[2]The Hong Kong University of Science & Technology,
[3]University of California, Berkeley, [4]PhiGent Robotics

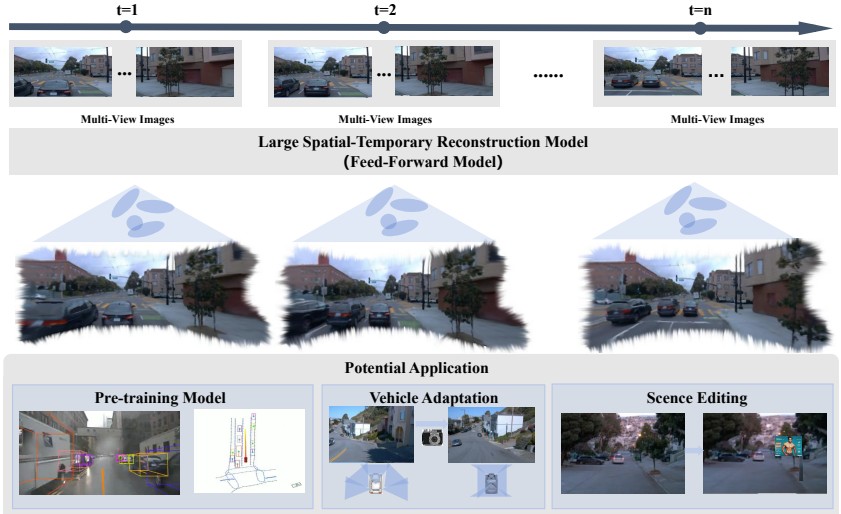

Figure 1: Leveraging temporal multi-view images, the Large 4D Gaussian Reconstruction Model (DrivingRecon) can predict 4D driving scenes. DrivingRecon serves as a pre-trained model that effectively captures geometric and motion information. Additionally, DrivingRecon can synthesize novel views based on specific camera parameters, ensuring adaptability to various vehicle models. DrivingRecon further facilitates editing of designated 4D scenes.

## Abstract

Large reconstruction model has remarkable progress, which can directly predict 3D or 4D representations for unseen scenes and objects. However, current work has not systematically explored the potential of large reconstruction models in the field of autonomous driving. To achieve this, we introduce the Large 4D Gaussian Reconstruction Model (DrivingRecon). With an elaborate and simple framework design, it not only ensures efficient and high-quality reconstruction, but also provides potential for downstream tasks. There are two core contributions: firstly, the Prune and Dilate Block (PD-Block) is proposed to prune redundant and overlapping Gaussian points and dilate Gaussian points for complex objects. Then, dynamic and static decoupling is tailored to better learn the temporary-consistent geometry across different time. Experimental results demonstrate that DrivingRecon significantly improves scene reconstruction quality compared to existing methods. Furthermore, we explore applications of DrivingRecon in model pre-training, vehicle type adaptation, and scene editing. Our code is available at DriveRecon.

---

[*]Work done while visiting UC Berkeley.
[†]Corresponding author.

39th Conference on Neural Information Processing Systems (NeurIPS 2025).

# 1 Introduction

Autonomous driving has made remarkable advancements in recent years, particularly in the areas of perception [25, 76, 21, 58, 54], prediction [17, 14, 26, 71, 61], and planning [10, 8, 9, 19, 73, 72, 33]. With the emergence of end-to-end autonomous driving systems that directly derive control signals from sensor data [18, 19, 22], conventional open-loop evaluations have become less effective [74]. Real-world closed-loop evaluations offer a promising solution, where the key lies in the development of high-quality scene reconstruction [50, 62, 30, 29, 31, 27, 28, 39, 69, 34].

Despite numerous advances in photorealistic reconstruction of small-scale scenes [37, 38, 4, 23, 57, 34], modeling large-scale and dynamic driving environments remains challenging. Most existing methods tackle these challenges by using 3D bounding boxes to differentiate static from dynamic components [64, 60, 50]. Subsequent methods learn the dynamics in a self-supervised manner with a 4D NeRF [66] or 3D Gaussian [20]. The aforementioned methods require numerous and time-consuming iterations for reconstruction and cannot generalize to new scenes.

Some recent methods are capable of reconstructing 3D objects [16, 75] or 3D indoor scenes [3, 6, 46] in a forward way. Many feed-forward approaches have also been tried in the autonomous driving [48, 42, 65, 56, 55]. However, all feed-forward methods predict a Gaussian point for each pixel, and the pixels from multi-view images are fused together. This paradigm has the following disadvantages: (1) The multi-view images have overlaps, and the Gaussian points corresponding to the overlapping regions will cause artifact. (2) Complex objects require more Gaussian points to represent, and the fixed number of predicted Gaussian points per region limits the quality of the image. Besides, only same moment images are used to supervise the rendered dynamic object, which limits the geometric and apparent quality of the dynamic object.

To this end, we introduce a Large Spatial-Temporal Reconstruction Model (DrivingRecon) for autonomous driving. Our framework uses the shared 2D encoder and range-view decoder, which ensures that the image encoder can serve multiple autonomous downstream tasks. The most important innovation to solve pain points is that the decoder is consisting of Prune and Dilate Blocks (PD-Blocks). The PD-Block effectively prunes overlapping Gaussian points between adjacent views and redundant background points. The pruned Gaussian points can be replaced by dilated Gaussian points of complex object. Finally, static and dynamic decoupling is proposed to improve the quality of dynamic objects. The experimental results show that our method is significantly improved compared with the existing algorithms. Our main contributions are as follows:

- We propose the PD-Block, which learns to prune redundant Gaussian points from different views and background regions. It also learns to dilate Gaussian points for complex objects, enhancing the quality of reconstruction.
- We design rendering strategies for both static and dynamic components, allowing rendered images to be efficiently supervised across temporal sequences.
- We explore the effectiveness of DrivingRecon in reconstruction, pre-training, vehicle adaptation, and scene editing tasks.

# 2 Related Work

## 2.1 Driving Scene Reconstruction

Numerous efforts have been put into reconstructing scenes from autonomous driving data captured in real scenes. Existing self-driving simulation engines such as CARLA [12] or AirSim [43] suffer from costly manual effort to create virtual environments and the lack of realism in the generated data. Many studies have investigated the application of these methods for reconstructing street scenes. Block-NeRF [70] and Mega-NeRF [49] propose segmenting scenes into distinct blocks for individual modeling. Urban Radiance Field [40] enhances NeRF training with geometric information from LiDAR, while DNMP [32] utilizes a pre-trained deformable mesh primitive to represent the scene. Streetsurf [15] divides scenes into close-range, distant-view, and sky categories, yielding superior reconstruction results for urban street surfaces. MARS [60] employs separate networks for modeling background and vehicles, establishing an instance-aware simulation framework. With the introduction of 3DGS [23], DrivingGaussian [77] introduces Composite Dynamic Gaussian Graphs and incremental static Gaussians, while StreetGaussian [64] optimizes the tracked pose of dynamic

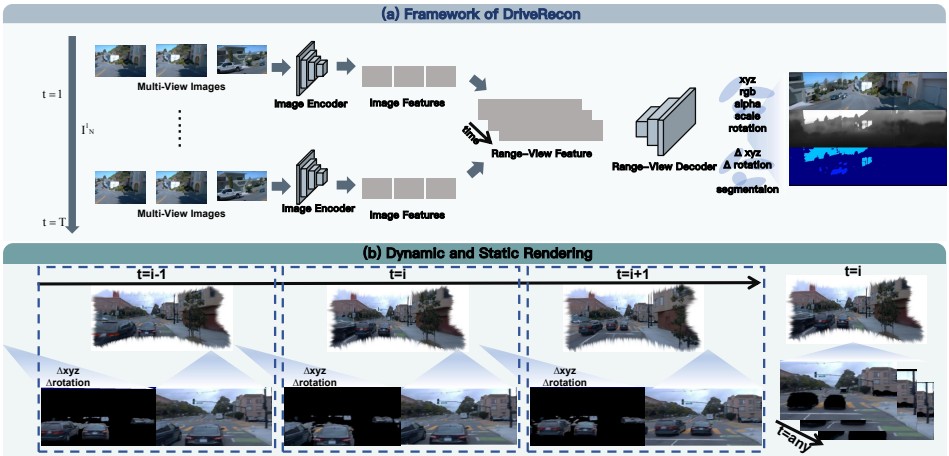

Figure 2: The overview of DrivingRecon. (a) Framework: multi-view images are in turn sent to the shared image encoder, and range view decoder to directly predict 4D Gaussians. (b) For dynamic objects, we only use next time-step images to supervise the current Gaussian parameters. For static scenes, rendering supervision is used across timestamps. In addition, reconstruction loss is also applied.

Gaussians and introduces 4D spherical harmonics for varying vehicle appearances across frames. Omnire [7] further focus on the modeling of non-rigid objects in driving scenarios. However, these reconstruction algorithms requires time-consuming iterations to build a new scene.

## 2.2 Large Reconstruction Models

Some works have proposed to greatly speed this up by training neural networks to directly learn the full reconstruction task in a way that generalizes to novel scenes [70, 52, 51, 59]. Recently, LRM [16] was among the first to utilize large-scale multiview datasets including Objaverse [11] to train a transformer-based model for NeRF reconstruction. The resulting model exhibits better generalization and higher quality reconstruction of object-centric 3D shapes from sparse posed images in a single model forward pass. Similar works have investigated changing the representation to Gaussian splatting [47, 65], introducing architectural changes to support higher resolution [63, 44], and extending the approach to 3D scenes [2, 6]. Recently, L4GM utilize temporal cross attention to fuses multiple frame information to predict the Gaussian representation of a dynamic object [41]. Many feed-forward approaches have also been tried in the autonomous driving [48, 42, 65, 56]. However, all feed-forward methods predict a Gaussian point for each pixel. This paradigm will cause artifacts in the overlapping areas of multiple view.

## 3 Method

In this section, we present the Large 4D Reconstruction Model (DrivingRecon), which generates 4D scenes from surround-view video inputs in a single feed-forward pass. Section 3.1 details the overview of DrivingRecon. In Section 3.2, we provide an in-depth examination of the Prune and Dilate Block (PD-Block). Finally, Section 3.3 discusses our training, which includes static and dynamic decoupling strategy.

### 3.1 Overall framework

**Problem Definition.** DrivingRecon utilizes temporal multi-view images $D$ to train a feed-forward model $\mathbf{G} = f(D, \mathcal{E}, \mathcal{R}, \mathcal{V})$. For the $i$-th sample, $D^i = \{X^t, K^t, E^t \mid t = 1, \ldots, T\}$ includes $N$ multi-view images $X^t = \{I_1, \ldots, I_j, \ldots, I_N\}$ at each timestep $t$, with corresponding intrinsic parameters $\mathcal{E}^t = \{E_1, \ldots, E_j, \ldots, E_N\}$, extrinsic rotation $\mathcal{R}^t = \{R_1, \ldots, R_j, \ldots, R_N\}$, and extrinsic translation $\mathcal{V}^t = \{V_1, \ldots, V_j, \ldots, V_N\}$. The extrinsic parameter is to project the camera coordinate system directly into the world coordinate system. We take the video start frame as the

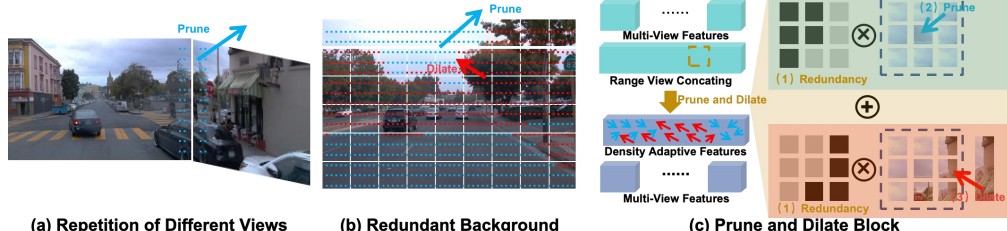

Figure 3: The motivation and details of Prune and Dilate Block (PD-Block). (a) Different views predict repeated Gaussian points, causing the model collapse. (b) Simple backgrounds (blue dots) do not need a large number of Gaussian dots to be represented, while complex objects (red dots) need more Gaussian dots to be represented. (c) PD-Block fuse the multi-view image features into a range view form. Then PD-Block prune and dilate the Gaussian points according to the complexity of the scene.

origin of the world coordinate system. This feed-forward model predicts Gaussians $\mathcal{G} = \{\mathbf{G} \in \mathbb{R}^d\}$ in the structure of $(\mathbf{xyz} \in \mathbb{R}^3, \mathbf{rgb} \in \mathbb{R}^3, \mathbf{a} \in \mathbb{R}^1, \mathbf{s} \in \mathbb{R}^3, \mathbf{c} \in \mathbb{R}^{|\mathcal{C}|}, \mathbf{r} \in \mathbb{R}^4, \mathbf{\Delta xyz} \in \mathbb{R}^3, \mathbf{\Delta r} \in \mathbb{R}^4)$. These elements represent position, RGB color, scale, rotation vectors, semantic logits, position change and rotation change, respectively.

**Motivation.** DriveRecon uses the shared 2D encoder and range-view decoder. The framework was designed with two principles in mind: (1) Pre-training. Driving tasks such as end-to-end and object detection typically use a shared image encoder [25, 76, 53, 22]. For 4D reconstruction tasks, the shared image encoder can learn geometry, motion, and appearance information. (2) Efficient fusion. Unlike perception and planning tasks, 4D reconstruction requires decoding high resolution features. The range-view decoder can splice adjacent image features together, so that features with similar space can be efficiently fused [24, 13].

**Pipeline.** Specifically, the temporal multi-view images $D$ are processed through a shared image encoder $F_{img}$ to extract image features $e_{img}$ as shown in Fig. 2. The 3D position code is stitched into the channel of the image. Adjacent image features are concatenated together in the range view form. Then, different time features are stitched together in different channels. These features are fed into the range-view decoder. The range-view decoder mades up of the Prune and Dilate block (PD-Block). Finally, the Gaussian adapter in range-view decoder transforms the decoded features into Gaussian points.

## 3.2 Learn to Prune and Dilate

The feed-forward method predict a Gaussian point for each pixel, which causes the following disadvantages: (1) The Gaussian points corresponding to the overlapping regions will cause artifact as shown in Fig 3 (a). (2) Complex objects require more Gaussian points to represent, and the fixed number of predicted Gaussian points per region limits the quality of the image as shown in Fig 3 (b). So, we propose the Prune and Dilate Block (PD-Block) that adaptively predicts the number of Gaussian points at different areas. It can prune Gaussian points on overlapping parts and dilate the number of Gaussian points on complex objects to improve rendering quality.

**Prune and Dilate Block.** We propose the Prune and Dilate Block (PD-Block), which can dilate the Gaussian point of complex instances and prune the redundant Gaussian as shown in Figure 2 (c). The core steps of PD-Block are: (1) **Redundancy definition**. Given range-view features, we evenly propose $K$ centers in space, and the center feature is computed by averaging its $Z$ nearest points [36]. We then calculate the pair-wise cosine similarity matrix $S$ between the region feature and the center points. (2) **Redundancy Pruning.** We set a threshold $\tau$ to generate a mask $M$ that is considered 0 if it is below this threshold and 1 if it is above this threshold. In addition, the point most similar to the center has always been retained. (3) **Complexity Dilating.** Based on mask, we can aggregate the long-term features $e_{lt}$ and the local features $e_{lc}$, $e = M * e_{lt} + (1 - M) * e_{lc}$. Here, the long-term features $e_{lt}$ is extracted by a large kernal convolution, and the local features $e_{lc}$ is the original range view features.

**Unaligned Gaussian Adapter.** The Gaussian adapter employs two convolutional blocks to convert features into segmentation $\mathbf{c} \in \mathbb{R}^C$, depth categories $\mathbf{d_c} \in \mathbb{R}^L$, depth regression refinement $\mathbf{d_r} \in \mathbb{R}^1$, RGB color $\mathbf{rgb} \in \mathbb{R}^3$, alpha $\mathbf{a} \in \mathbb{R}^1$, scale $\mathbf{r} \in \mathbb{R}^3$, rotation $\mathbf{r} \in \mathbb{R}^3$, UV-coordinate shifts $[\Delta u, \Delta v]$, and optical flow $[\Delta x, \Delta y, \Delta z]$. The activation functions for RGB color, alpha, scale, and rotation are consistent with those in [47]. The final depth per pixel is computed as $\mathbf{d_f} = \sum_{l=1}^{L} l \times \text{softmax}(\mathbf{d_c}) + \mathbf{d_r}$.

However, PD-Blocks effectively manage spatial computational redundancy by reallocating resources from simple scenes to more complex objects, allowing for Gaussian points that are not strictly pixel-aligned. For this reason, our Gaussian Adapter also predicts the offset of the uv coordinate $[\Delta u, \Delta v]$. The world coordinate $[x, y, z] = RE^{-1}\mathbf{d_f} * [u + \Delta u, v + \Delta v, 1] + V$. The above operations are universal for any time and view, so we did not label the time and views for simplicity. Gaussian points of different viewing angles are all fused to render. In addition, we can use the world coordinates at time t and the predicted optical flow to get the world coordinates at time t+1, $[x_{t+1}, y_{t+1}, z_{t+1}] = [x_t + \Delta x_t, y_t + \Delta y_t, z_t + \Delta z_t]$. Rotational changes in an object are interpreted as positional changes.

### 3.3 Static and Dynamic Decoupling

To learn geometry and motion information, DrivingRecon carefully designed the static and dynamic decoupling as shown in Fig. 2, including both unsupervised and supervised manner.

**Unsupervised manner.** The views of the driving scene are very sparse, meaning that only a limited number of cameras capture the same scene simultaneously. Hence, cross-temporal view supervision is essential. For dynamic objects, our algorithm predicts not only the current Gaussian of dynamic objects at time $t$ but also predicts the motion flow of each Gaussian point. Therefore, we will also use the next frame to supervise the predicted Gaussian points, i.e., dynamic reconstruction loss $\mathcal{L}_{dy}$. Additionally, we have the reconstruction constraint $\mathcal{L}_{re}$, which involves rendering the image as the same as the input. The dynamic and reconstruction constraint all use the L1 constraint to constitute an unsupervised constraint $\mathcal{L}_{re}$. We also employ cross-entropy loss $\mathcal{L}_c$ for the depth categories $_c$ predicted by the Gaussian Adapter and L1 loss $\mathcal{L}_r$ for the refined depth $_r$. In the field of autonomous driving, depth supervision during training is considered unsupervised [65, 20, 66].

**Supervised manner.** For autonomous driving, we also want to know the semantic properties of each Gaussian point in 4D space. So, cross-entropy loss is used to constrain the segmentation results predicted by Gaussian Adapter, i.e., $\mathcal{L}_{seg}$. Segmentation labels can be provided by some existing segmentation models, but we still consider them supervised for fair comparison. With the segmentation model, we can render the scene with camera parameters of adjacent timestamps and supervise only the static part, i.e., static reconstruction loss $\mathcal{L}_{dy}$. It is important to note that when supervising the rendering across the time sequence, we will not supervise the rendered image where the threshold value is less than $\alpha$, as these pixels often do not overlap across the time sequence. In summary, the overall constraints for training DrivingRecon are:

$$\mathcal{L}_{\text{total}} = \underbrace{\lambda_c \mathcal{L}_c + \lambda_r \mathcal{L}_r + \lambda_{re} \mathcal{L}_{re} + \lambda_{dr} \mathcal{L}_{dr}}_{\text{Unsupervised}}$$
$$+ \underbrace{\lambda_{sr} \mathcal{L}_{sr} + \lambda_{seg} \mathcal{L}_{seg}}_{\text{Supervised}}$$

where each $\lambda$ term balances the contribution of the respective loss component. $\mathcal{L}_{sr}$ and $\mathcal{L}_{seg}$ used segmentation labels, which is not used for pre-training experiment. Other loss are considered unsupervised, which also allows DrivingRecon to achieve good performance. These collective regulations and constraints enable DrivingRecon to effectively integrate geometry and motion information, enhancing its capacity for accurate scene reconstruction across time and perspectives.

## 4 Experiment

In this section, we evaluate the performance of DrivingRecon in terms of reconstruction and novel view synthesis, as well as explore its potential applications. We also provide detailed information on the dataset setup, baseline methods, and implementation details.

Table 1: **Comparison to state-of-the-art methods on the Waymo Open Dataset.** PSNR, SSIM, and Depth RMSE (D-RMSE) are reported. Speed metrics are estimated on a single A100 GPU. *: reproduced by us. †: Non-sky region. ⁻: not using segmentation supervision.

| Methods | Dynamic-only | | | Full image† | | | Inference speed | Real-time rendering |
| | PSNR↑ | SSIM↑ | D-RMSE↓ | PSNR↑ | SSIM↑ | D-RMSE↓ | Time↓ | (>200FPS) |
|---|---|---|---|---|---|---|---|---|
| *Per-Scene Optimization methods* | | | | | | | | |
| EmerNeRF [66] | 17.79 | 0.255 | 40.88 | 24.51 | 0.738 | 33.99 | 14min | × |
| 3DGS [23] | 17.13 | 0.267 | 13.88 | 25.13 | 0.741 | 19.68 | 23min | ✓ |
| PVG [5] | 15.51 | 0.128 | 15.91 | 22.38 | 0.661 | 13.01 | 27min | ✓ |
| DeformableGS [68] | 17.10 | 0.266 | 12.14 | 25.29 | 0.761 | 14.79 | 29min | ✓ |
| *Generalizable feed-forward methods* | | | | | | | | |
| LGM [47] | 19.58 | 0.443 | 9.43 | 23.59 | 0.691 | 8.02 | 0.06s | ✓ |
| GS-LRM [75] | 20.02 | 0.520 | 9.95 | 25.18 | 0.753 | 7.94 | **0.02s** | ✓ |
| SCube* [42] | 20.51 | 0.532 | 8.82 | 25.72 | 0.783 | 5.62 | 0.42s | ✓ |
| DrivingForward* [48] | 21.97 | 0.622 | 7.42 | 26.32 | 0.774 | 5.79 | 0.21s | ✓ |
| STORM [65] | 22.10 | 0.624 | 7.50 | 26.38 | 0.794 | 5.48 | 0.18s | ✓ |
| *Ours* | | | | | | | | |
| DriveRecon⁻ | 23.11 | 0.641 | 7.32 | 27.23 | 0.819 | 5.23 | 0.08s | ✓ |
| DriveRecon | **24.01** | **0.662** | **7.03** | **27.52** | **0.833** | **5.14** | 0.08s | ✓ |

## 4.1 Datasets.

Following [48, 42, 65, 56], we use both the Waymo Open dataset [45] and the nuScenes [1] to test the algorithm's performance. The rendered image resolution of DrivingRecon is $160 \times 240$ as same as [65]. For nuScenes dataset, the rendered image resolution of DrivingRecon is $224 \times 400$ as same as [56]. All training and testing validation are consistent with the official ones.

## 4.2 Training Details.

The model is trained on 24 NVIDIA A100 (80G) GPUs for 50000 iterations, all about 24 gpu days. A batch size of 2 for each GPU is used under bfloat16 precision, resulting in an effective batch size of 48. The AdamW optimizer is employed with a learning rate of $4 * 10^{-4}$ and a weight decay of 0.05. $\lambda_{re}, \lambda_c, \lambda_r, \lambda_{dr}, \lambda_{sr}, \lambda_{seg}$ are set as 1.0, 0.1, 0.1, 0.1, 0.1, 0.1, respectively. Following [65], The input to our model consists of 4 context timesteps, evenly spaced at $t + 0s$, $t + 0.5s$, $t + 1.0s$, and $t + 1.5s$, where $t$ is a randomly chosen starting timestep. These balance parameters are based on our experience.

## 4.3 Main Results

We compare our method against both two categories of approaches: per-scene optimization methods and feed-forward models. For per-scene optimization, we evaluate against both NeRF-based and 3DGS-based approach, including EmerNeRF [66], 3DGS [23], PVG [5], and DeformableGS [68]. Since LiDAR data is not provided at test time in our setup, these baselines is also without LiDAR supervision to ensure a fair comparison. In the second category, we compare against recent large reconstruction models, including LGM [47], GS-LRM [75], SCube [42], DrivingForward [48], STORM [65]. SCube and DrivingForward were modified to the same resolution and test protocol using their officially provided code.

We present the quantitative results in Tab. 1. Compared to per-scene optimization methods, the large reconstruction models achieves comparable performance in both dynamic regions and full images in terms of photorealism, geometry, and inference speed. Notably, compared to other generalizable feed-forward models, DriveRecon demonstrates a robust ability to model scene dynamics and process multi-timestep, multi-view images. DriveRecon also achieves the best performance when it has no segmentation supervision, that is, pure unsupervised. In fact, both Scube and STORM use different forms of segmentation supervision. Besides, DriveRecon achieves these improvements while reducing inference time to just 0.08 second. The sota method STORM has 100.60M parameters, and DriveRecon only has 53.37M parameters. This is because we used the shared image encoder and the range view decoder. It ensures that our model parameters and reasoning are significantly efficient.

To better clarify the effectiveness of our algorithm. We compare Scube and OmniScene, two new generalizable Gaussian methods. STORM is not open source so it is not compared. As shown in Fig. 4,

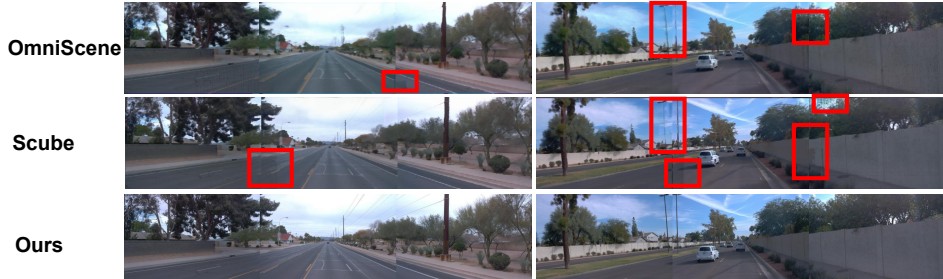

Figure 4: Reconstruction quantitative comparison. The red box indicates the presence of artifacts in the overlap area of multi-view images, due to Gaussian prediction at per-pixel level **(zoom in for the best view.)**

Scube and OmniScene have serious artifacts in overlapping areas. This is because these method predict a Gaussian point for each pixel, and the same location in the overlapping region may correspond to multiple Gaussian points. Our PD-Block can dynamically remove these redundant points, and the range-view decoder can better integrate different views. This responds to the motivation of our paper.

Table 2: Comparison to state-of-the-art methods on the nuScenes. $^*$: reproduced by us. Pearson Correlation Coefficient (PCC) quantifies the statistical relationship between the predicted depth and ground-truth depth as the scale-invariant metric [56].

| Method | PSNR↑ | SSIM↑ | LPIPS↓ | PCC↑ |
|---|---|---|---|---|
| pixelSplat [3] | 21.51 | 0.616 | 0.372 | 0.001 |
| MVSplat [6] | 21.61 | 0.658 | 0.295 | 0.181 |
| DrivingForward$^*$ [48] | 24.32 | 0.732 | 0.229 | 0.766 |
| Omni-Scene [56] | 24.27 | 0.736 | 0.237 | 0.804 |
| DriveRecon | **25.42** | **0.782** | **0.201** | **0.825** |

Further, we compare recent DrivingForward [48] and Omni-Scene [56] simultaneously on nuScenes. The DrivingForward is reproduced using the same protocol as [56]. STORM is not open source, and Scube relies heavily on more fine-grained Occ. So, we can't reproduce these method on nuScenes. As shown in Tab. 2, we have also demonstrated the advance of our approach. This proves that our PD-Block and dynamic and static decoupling are very efficient.

## 4.4 Ablation study

To assess the effectiveness of our proposed algorithm, we conducted a series of ablation experiments. The key components under evaluation include the PD-Block and Dynamic and Static Decoupling(DS).

As shown in Tab. 3, each module contributes significant performance improvements. Notably, the PD-Block achieves the highest enhancement. This improvement stems from two primary factors: (1) an optimized distribution of computational resources based on spatial complexity, where more Gaussian points are allocated to complex regions while simpler backgrounds receive fewer points; (2) enhanced multi-view integra-

Table 3: Ablation of DrivingRecon.

| | PSNR | SSIM | LPIPS |
|---|---|---|---|
| all | 27.52 | 0.83 | 0.16 |
| w/o PD-Block | 26.21 | 0.78 | 0.24 |
| w/o DS | 26.44 | 0.79 | 0.20 |

tion within a broad field of view. The DS-R mechanism also led to marked improvements, largely attributed to the use of cross-temporal supervision for better dynamic and static object differentiation. Qualitative visualization of the new view can be seen in Fig. 5. It proved that we can render geometrically consistent new perspectives in different time.

## 4.5 Potential application

**Vehicle adaptation.** The introduction of a new car model may result in changes in camera parameters, such as camera type (intrinsic parameters) and camera placement (extrinsic parameters) [53, 35].

**Different rendering views**

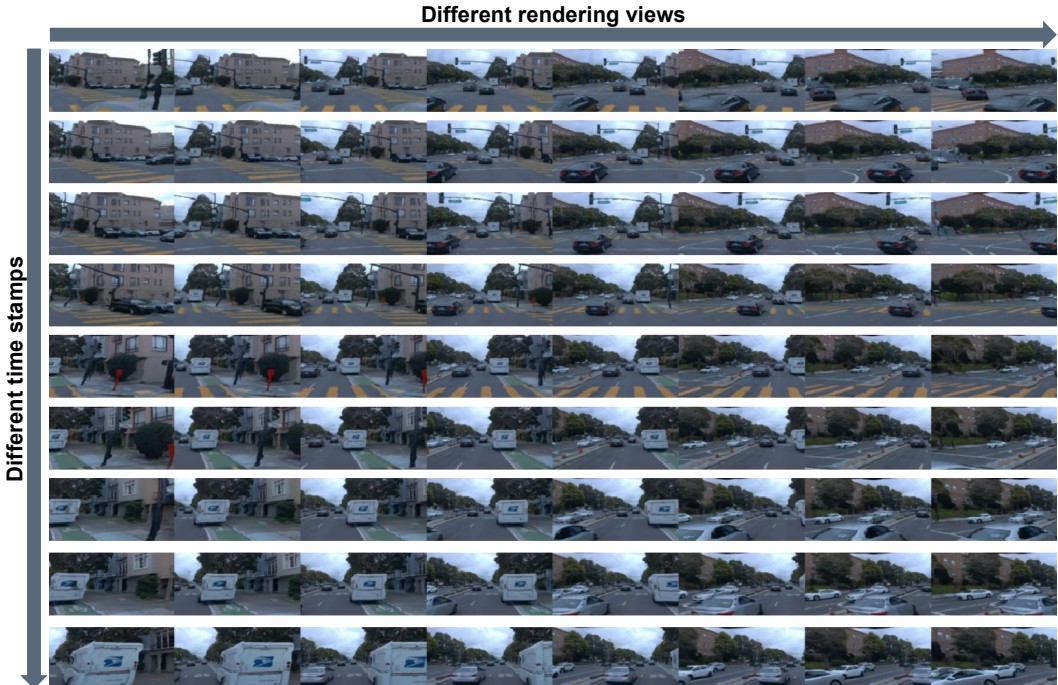

Figure 5: Novel view rendering. Based on the predicted Gaussians, we render different views at different times. The novel views are of very high quality and very high spatio-temporal consistency **(zoom in for the best view.)**

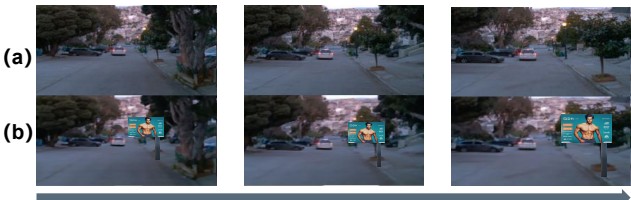

Figure 6: Scene editing. We can insert the new object in the scene, and ensure time consistency.

Table 4: Comparison of different approaches on domain generalization protocols, where * stands for using aligned intrinsic parameters, + stands for randomly augmenting camera extrinsic parameters.

| Waymo → nuScenes | Target Domain (nuScenes) | | | | |
|---|---|---|---|---|---|
| Method | mAP↑ | mATE↓ | mASE↓ | mAOE↓ | NDS* ↑ |
| Oracle | 0.475 | 0.577 | 0.177 | 0.147 | 0.587 |
| DG-BEV | 0.303 | 0.689 | 0.218 | 0.171 | 0.472 |
| PD-BEV | 0.311 | 0.686 | 0.216 | 0.170 | 0.478 |
| Ours* | 0.305 | 0.690 | 0.219 | 0.167 | 0.471 |
| **Ours*+** | **0.323** | **0.675** | **0.212** | **0.166** | **0.490** |

The 4D reconstruction model is capable of rendering images with different camera parameters to mitigate the potential overfitting of these parameters. To achieve this, we rendered images on Waymo with random intrinsic parameters and performed random rendering of novel views as a form of data augmentation. Based on PD-BEV[3], we used this jointly rendered and original data to train the BEVDepth on Waymo, following the approach of [53, 35]. It is important to note that our rendered images also undergo an augmentation pipeline as part of the detection algorithm, including resizing and cropping.

---

[3] `https://github.com/EnVision-Research/Generalizable-BEV`

Table 5: Performance gain of our method for joint perception, prediction, and planning.

| Method | Detection | | Tracking | | | Future Occupancy Prediction | | | |
| --- | --- | --- | --- | --- | --- | --- | --- | --- | --- |
| | NDS ↑ | mAP ↑ | AMOTA↑ | AMOTP↓ | IDS↓ | IoU-n.↑ | IoU-f.↑ | VPQ-n.↑ | VPQ-f.↑ |
| UniAD | 49.36 | 37.96 | 38.3 | 1.32 | 1054 | 62.8 | 40.1 | 54.6 | 33.9 |
| ViDAR | 52.57 | 42.33 | 42.0 | 1.25 | 991 | 65.4 | 42.1 | 57.3 | 36.4 |
| **Ours+** | **53.21** | **43.21** | **42.9** | **1.18** | **948** | **66.5** | **43.3** | **58.2** | **37.3** |

| Method | Mapping | | Motion Forecasting | | | Planning | |
| --- | --- | --- | --- | --- | --- | --- | --- |
| | IoU-lane↑ | IoU-road↑ | minADE↓ | minFDE↓ | MR↓ | avg.L2↓ | avg.Col.↓ |
| UniAD | 31.3 | 69.1 | 0.75 | 1.08 | 0.158 | 1.12 | 0.27 |
| ViDAR | 33.2 | 71.4 | 0.67 | 0.99 | 0.149 | 0.91 | 0.23 |
| **Ours+** | **33.9** | **72.1** | **0.60** | **0.89** | **0.138** | **0.84** | **0.19** |

As demonstrated in Tab. 4, when we employ both camera intrinsic and extrinsic parameter augmentation, we observe a significant improvement in performance. However, the use of only camera intrinsic parameter augmentation did not yield good results, due to the superior ability of virtual depth in addressing the issue of camera intrinsic parameters. The utilization of multiple extrinsic parameters helps the algorithm learn the stereo relationship between cameras more effectively.

**Pre-training model.** The 4D reconstruction network is capable of understanding the geometric information of the scene, the motion trajectory of dynamic objects, and the semantic information. These capabilities are reflected in the encoding of images, where the weights of these encoders are shared. We replaced our encoder with the ResNet-50, which is a commonly used base network for many algorithms as same as [67]. Following VIDAR [4], we then retrained the 4D reconstruction network using this configuration, without using any segmentation annotations, resulting in completely unsupervised pre-training. Subsequently, we replaced the encoder of UniAD [19] with our pre-trained model and fine-tuned it on the nuScenes dataset. The results, as presented in Tab. 5, demonstrate that our pre-trained model achieved better performance compared to ViDAR [67], highlighting the ability of our algorithm to leverage large-scale unsupervised data for pre-training and improving multiple downstream tasks.

**Scene editing.** The 4D scene reconstruction model enables us to obtain comprehensive 4D geometry information of a scene, which allows for the removal, insertion, and control of objects within the scene. As shown in Fig. 6, we added billboards with people's faces to fixed positions in the scene, representing a corner case where cars come to a stop. As can be seen from the figure, the scenario we created exhibits a high level of temporal consistency.

## 5  Conclusion

The paper introduces DrivingRecon, a novel 4D Gaussian Reconstruction Model for fast 4D reconstruction of driving scenes using surround-view video inputs. The entire framework uses a shared image encoder and range view decoder. A key innovation is the Prune and Dilate Block (PD-Block), which prunes redundant Gaussian points from adjacent views and dilates points around complex edges, enhancing the reconstruction of dynamic and static objects. Additionally, a dynamic-static rendering approach using optical flow prediction allows for better supervision of moving objects across time sequences. DrivingRecon shows superior performance in scene reconstruction and novel view synthesis compared to existing methods. It is particularly effective for tasks such as model pre-training, vehicle adaptation, and scene editing. We quantitatively evaluated pre-training and vehicle adaptation and made significant improvements.

## 6  Acknowledgments

This work is supported by National Natural Science Foundation of China (No. 62206068), HKUST-HKUST(GZ) Cross-Campus Collaborative Research Scheme (Project No. C036) and Guangdong Provincial Department of Science and Technology's '1+1+1' Joint Funding Program for Guangdong-Hong Kong Universities.

---

[4] https://github.com/OpenDriveLab/ViDAR

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
